# Ca^2+^ Influx through TRPC Channels Is Regulated by Homocysteine–Copper Complexes

**DOI:** 10.3390/biom13060952

**Published:** 2023-06-06

**Authors:** Gui-Lan Chen, Bo Zeng, Hongni Jiang, Nikoleta Daskoulidou, Rahul Saurabh, Rumbidzai J. Chitando, Shang-Zhong Xu

**Affiliations:** 1Centre for Atherothrombosis and Metabolic Disease, Hull York Medical School, University of Hull, Hull HU6 7RX, UK; chenguilan@swmu.edu.cn (G.-L.C.); zengbo@swmu.edu.cn (B.Z.);; 2Diabetes, Endocrinology and Metabolism, Hull York Medical School, University of Hull, Hull HU6 7RX, UK

**Keywords:** homocysteine, calcium channel, TRPC, TRPM2, copper, endothelial cells, angiogenesis, 2-aminoethoxydiphenyl borate

## Abstract

An elevated level of circulating homocysteine (Hcy) has been regarded as an independent risk factor for cardiovascular disease; however, the clinical benefit of Hcy lowering-therapy is not satisfying. To explore potential unrevealed mechanisms, we investigated the roles of Ca^2+^ influx through TRPC channels and regulation by Hcy–copper complexes. Using primary cultured human aortic endothelial cells and HEK-293 T-REx cells with inducible TRPC gene expression, we found that Hcy increased the Ca^2+^ influx in vascular endothelial cells through the activation of TRPC4 and TRPC5. The activity of TRPC4 and TRPC5 was regulated by extracellular divalent copper (Cu^2+^) and Hcy. Hcy prevented channel activation by divalent copper, but monovalent copper (Cu^+^) had no effect on the TRPC channels. The glutamic acids (E542/E543) and the cysteine residue (C554) in the extracellular pore region of the TRPC4 channel mediated the effect of Hcy–copper complexes. The interaction of Hcy–copper significantly regulated endothelial proliferation, migration, and angiogenesis. Our results suggest that Hcy–copper complexes function as a new pair of endogenous regulators for TRPC channel activity. This finding gives a new understanding of the pathogenesis of hyperhomocysteinemia and may explain the unsatisfying clinical outcome of Hcy-lowering therapy and the potential benefit of copper-chelating therapy.

## 1. Introduction

Cardiovascular disease (CVD) is the leading cause of death in developed nations and is increasing rapidly in developing countries. The well-described risk factors include high blood pressure, dyslipidemia, smoking, diabetes mellitus, obesity, and new independent risk factors, such as C-reactive protein, lipoprotein (a), fibrinogen, and homocysteine (Hcy). The association between elevated Hcy levels and atherosclerosis was first demonstrated in patients with hyperhomocysteinemia in 1969 [1]; however, the importance of Hcy as a risk factor has been especially acknowledged during the last two decades in that even a mild or moderate increase in Hcy level (>15 µmol/L) in serum or plasma is closely associated with the morbidity and mortality of coronary heart diseases [2,3,4,5,6], stroke [7,8], peripheral vascular disease [9], venous thrombosis [10], dementia or Alzheimer’s disease [11], nerve degeneration [12], diabetes [13], osteoporotic fractures [14], end-stage renal disease [15], and other conditions, such as adverse pregnancy outcome (early abortion, placental vasculopathy, and birth defects) [16] and liver fibrosis [17]. In patients with genetic enzyme defects including cystathionine β-synthase (CBS), methylenetetrahydrofolate reductase (MTHFR), and methionine synthase (MS) in the Hcy metabolic pathway, the concentration of Hcy is much higher and accompanied with more severe cardiovascular damage [8,18]. The MTHFR (T677C point mutation) variant is the most common enzyme defect associated with high Hcy and its prevalence is 5~15% in Caucasian and Asian populations. The mechanisms of how Hcy causes diseases or becomes a risk for diseases are still unknown [19,20]; in particular, the intervention for lowering plasma Hcy levels in patients did not show any preventive effects against cardiovascular diseases [21,22], suggesting unrecognised mechanisms or interactions with Hcy may exist in vivo. Since Hcy is involved in the pathogenesis of many diseases and is associated with all-cause mortality [23], it is reasonable to hypothesise that Hcy may target some ubiquitously expressed proteins or key signalling molecules in the body.

Calcium is a key signalling messenger in the cell and several studies have suggested that Hcy may interfere with Ca^2+^ signalling pathways. For example, Ca^2+^ influx and intracellular Ca^2+^ release were enhanced by Hcy [24], and the ligand-gated Ca^2+^ channel NMDA receptor was stimulated by Hcy [25]. Interestingly, it has been shown that the up-regulation of Ca^2+^ permeable channels, such as TRPC1 and TRPC5, is related to vascular neointimal growth and cell mobility [26,27], while neointimal growth was also observed in the blood vessels from patients with hyperhomocysteinemia [1]. TRPC channels are ubiquitously expressed in the cardiovascular system and mediate the common pathway of Ca^2+^ entry via G-protein coupled receptor activation and/or the depletion of the endoplasmic reticulum (ER) Ca^2+^ store [28,29]. Therefore, we hypothesised that TRPC channels could be involved in the pathophysiology of hyperhomocysteinemia. On the other hand, the correlation between Hcy and copper in cardiovascular disease has been demonstrated in clinical surveys [30,31,32,33], and copper-lowering therapy with a chelator could be beneficial for cardiac hypertrophy [34]. We, therefore, aimed to investigate the effects of Hcy on TRPC channels and its regulatory mechanisms with copper ions in causing endothelial dysfunction and subsequent atherogenicity.

## 2. Materials and Methods

### 2.1. Cell Culture and Transfection

Human TRPC4α (NM_016179), TRPC4β1 (NM_001135955, but the β1 isoform was cloned from the endothelial cell with one glutamic acid deletion at E785), and TRPC5 (AF054568) in the tetracycline-regulatory vector pcDNA4/TO (Invitrogen, Paisley, UK) were transfected into HEK-293 T-REx cells using the Lipofectamine^TM^ 2000 transfection reagent (Invitrogen, Paisley, UK). TRPC4 was tagged with an enhanced yellow fluorescent protein (EYFP) at the N-terminus. Expression was induced by 1 µg∙mL^−1^ tetracycline for 48–72 h before recording. The non-induced cells without the addition of tetracycline were used as a control. Cells were grown in DMEM-F12 medium (Invitrogen, Paisley, UK) containing 10% foetal calf serum (FCS), 100 units∙mL^−1^ penicillin, and 100 µg∙mL^−1^ streptomycin at 37 °C under 95% air and 5% CO_2_. Cells were seeded on coverslips prior to experiments.

Human aortic endothelial cells (HAECs) were purchased from PromoCell (Heidelberg, Germany) and cultured in an endothelial cell growth medium as we described previously [35,36]. The medium was supplemented with 2% foetal calf serum, 5.0 μg·L^−1^ epidermal growth factor, 0.5 μg·L^−1^ vascular endothelial growth factor, 10 μg·L^−1^ basic fibroblast factor, 20 μg·L^−1^ R3 IGF-1, and 22.5 mg·L^−1^ heparin. Cells in passages 2 to 4 were used in the experiment to avoid age-dependent variations.

### 2.2. Electrophysiological Recordings and Ca^2+^ Measurements

A whole-cell clamp was performed at room temperature (23–26 °C) as described before [37,38]. Briefly, the signal was amplified with an Axopatch B200 amplifier and controlled with pClamp software 10. A 1 s ramp voltage protocol from −100 mV to +100 mV was applied at a frequency of 0.2 Hz from a holding potential of 0 mV. Signals were sampled at 3 kHz and filtered at 1 kHz. A glass microelectrode with a resistance of 3–5 MΩ was used. The 200 nM Ca^2+^ buffered pipette solution contained 115 CsCl, 10 EGTA, 2 MgCl_2_, 10 HEPES, and 5.7 CaCl_2_ in mM. The pH was adjusted to 7.2 with CsOH and the osmolarity was adjusted to ~290 mOsm with mannitol. The calculated free Ca^2+^ was 200 nM using EQCAL (Biosoft, Cambridge, UK). The standard bath solution contained (mM): 130 NaCl, 5 KCl, 8 D-glucose, 10 HEPES, 1.2 MgCl_2,_ and 1.5 CaCl_2_. The pH was adjusted to 7.4 with NaOH. For excised patch recordings, the procedures were similar to our previous reports [39,40].

Intracellular Ca^2+^ was measured using a cuvette-based system as we described previously [35,41]. Briefly, HAECs were loaded with Fluo3-AM (5 µM) in a Ca^2+^ free standard bath solution (130 NaCl, 5 KCl, 8 D-glucose, 10 HEPES, and 1.2 MgCl_2_ in mM), then washed and resuspended in the standard bath solution. A total volume of 2 mL of standard bath solution with suspended cells was pipetted into a cuvette and the fluorescence was measured using a Perkin–Elmer LS50B fluorimeter. All electrophysiological recordings and Ca^2+^ measurements were performed at room temperature (25 °C).

### 2.3. RT-PCR

Total RNA was extracted from the cultured endothelial cells using TRI Reagent (Sigma-Aldrich, Poole, UK) and reverse transcribed with the Moloney murine leukaemia virus (M-MLV) reverse transcriptase using random primers (Promega, Southampton, UK). The PCR primer sequences used in this study and the detailed procedures were described in our previous report [42]. PCR products were confirmed by 2% agarose gel electrophoresis or direct sequencing.

### 2.4. Cell Proliferation, Migration, and Angiogenesis Assays

Endothelial cells were grown to confluence in 24-well plates in an endothelial cell medium. Cell proliferation was assayed by a WST-1 kit (Roche) as we reported [42,43]. For the cell migration assay, a linear scrape of ~0.3 mm width was made through a pipette tip [26]. The cells were cultured in an endothelial cell medium with or without Hcy. After 24 h of culture, the cells were fixed with 4% paraformaldehyde, and cells across the edge of the wound were counted. For the angiogenesis experiment, bovine skin collagen (Sigma, Hertfordshire, UK) was diluted to 1.5 mg/mL with extracellular matrix (ECM) (Sigma) at 2–8 °C as a working solution. The pH and osmolarity were adjusted by 1 M NaOH and 10× phosphate-buffered saline, respectively. Human vascular endothelial growth factor (Sigma, UK) was added to a final concentration of 20 ng/mL. Collagen working solution at a volume of 120 μL was added to each well of a 48-well plate and allowed to gelatinise for 30 min at 37 °C. EA.hy926 cells were resuspended in the ECM solution and added to each well at a volume of 300 μL (~3 × 10^4^ cells/well) and incubated at 37 °C for 30 min under 95% air and 5% CO_2_. After 24 h of culture with Hcy or the vehicle, cells were fixed with 4% paraformaldehyde, stained with 0.025% crystal violet, and photographed. The angiogenesis score was calculated by a semi-quantitative method as reported previously [44]. The BD Matrigel^TM^ (BD Bioscience, Chester, UK) was also used to see the effects of Hcy and Cu^2+^ on endothelial cell tube formation. The angiogenesis was analysed with Wim Tube software (Wimasis, Munich, Germany).

### 2.5. Reagents and Drugs

All general salts and reagents were purchased from Sigma-Aldrich (Poole, UK). L-homocysteine, lanthanum chloride (La^3+^), CuSO_4_ (Cu^2+^), gadolinium chloride (Gd^3+^), 2-aminoethoxydiphenyl borate (2-APB), trypsin, thapsigargin (TG), D-(−)-2-amino-5-phosphonopentanoic acid (D-AP5), verapamil, A23187, (1,10-phenanthroline)bis(triphenylphosphine)copper(I) nitrate dichloromethane adduct, and foetal calf serum were purchased from Sigma-Aldrich. Matrigel was purchased from BD Biosciences (UK) and Fluo-3 AM from Invitrogen (Paisley, UK). Fluo-3 AM (5 mM), TG (1 mM), and 2-APB (100 mM) were made up as stock solutions in 100% dimethyl sulphoxide (DMSO).

### 2.6. Statistics

Data are expressed as mean ± s.e.m. where *n* is the cell number for electrophysiological recordings and Ca^2+^ imaging. Data sets were compared using a paired *t*-test for the results before and after treatment, or the ANOVA Bonferroni’s post-hoc analysis for comparing more than two groups with significance indicated if *p* < 0.05.

## 3. Results

### 3.1. Ca^2+^ Influx Induced by Hcy in HAECs

The effect of Hcy on Ca^2+^ influx was measured in the primary cultured HAECs using Fluo-3 AM Ca^2+^ dye. Hcy at 1–100 µM increased the intracellular [Ca^2+^]_i_, which accounted for 33.1 ± 1.1% of the amplitude of the Ca^2+^ signal induced by calcium ionophore A231872 (Figure 1A,B). Blocking the voltage-gated Ca^2+^ channels with verapamil or using 100 mM K^+^ in the bath solution (equal molar substitution of Na^+^) to clamp the membrane potential did not prevent the effect of Hcy (Figure 1C,D), suggesting that Hcy-induced Ca^2+^ increase is mediated by non-voltage gated Ca^2+^-permeable channels. We also examined the Ca^2+^ release using the sarco/endoplasmic reticulum Ca^2^⁺-ATPase (SERCA) inhibitor thapsigargin (TG). Depletion of the ER Ca^2+^ store showed no significant blocking effect on Hcy-induced intracellular Ca^2+^ increase (Figure 1E). Hcy has been reported to induce Ca^2+^ transient through NMDA receptor activation in cultured neurons [24], therefore, we tested the effect of Hcy in cells treated with the NMDA antagonist D-(−)-2-amino-5-phosphonopentanoic acid (D-AP5). D-AP5 at 50 µM was unable to prevent the Hcy-induced Ca^2+^ influx (Figure 1F), suggesting that other Ca^2+^ entry pathways exist in endothelial cells. These results suggest that Hcy increases Ca^2+^ influx mainly through non-voltage gated channels, rather than the Ca^2+^ release or NMDA receptors in vascular endothelial cells.

### 3.2. Hcy-Induced Ca^2+^ Influx through TRPC4 and TRPC5 Channels

To explore which pathway is involved in Hcy-induced Ca^2+^ entry, we examined the expression and function of TRPC channels in endothelial cells. The mRNAs of TRPC1, 3, 4, and 6 were detected in the HAECs using RT-PCR. TRPC1 and TRPC4 were more abundant in HUVEC, but TRPC5 was low and TRPC3, TRPC6, and TRPC7 seemed to be absent in HUVEC (Figure 2A). The spliced isoforms of TRPC1^E9del^, TRPC4β1, and TRPC4Ɛ1 were also identified in the HAECs using the primer sets we reported previously [42] (Figure 2B).

Using whole-cell patch recordings, the effects of Hcy on TRPC4 and TRPC5 currents were examined in the HEK293 T-REx cells inducibly expressing TRPC channels [38]. Lanthanides (La^3+^ or Gd^3+^) were used as channel activators in our experiment as we used before [41,45]. After perfusion with Hcy, the currents of TRPC4 and TRPC5 were significantly stimulated (Figure 2C,D) while no effects were observed on the non-induced cells (Figure 2E,F), suggesting that Hcy induced Ca^2+^ influx via the activation of TRPC4 and TRPC5 channels.

### 3.3. Activation of TRPC4 and TRPC5 by Divalent Cu^2+^ and the Interference by Hcy

Hcy and copper are two important regulators of cellular oxidative stress and both are involved in atherogenicity, however, their mechanisms are unclear [30]. We found that divalent Cu^2+^ showed an initial transient inhibition and then a gradual activation of TRPC4α and TRPC5 currents after perfusion with 10 µM Cu^2+^ (Figure 3A,B). The current of TRPC4β1 was also activated by Cu^2+^ (Appendix A). The EC_50_ of Cu^2+^ for TRPC4α channel activation was 6.8 µM (Appendix A). The Cu^2+^-induced currents were also sensitive to the non-selective TRPC blocker 2-APB as the currents of TRPC4 and TRPC5 induced by lanthanides [41,45]. Interestingly, perfusion with Hcy (100 µM) completely prevented the TRPC4 and TRPC5 channel activation by Cu^2+^ (Figure 3C,D), suggesting that the interaction of Hcy and copper is critical for regulating TRPC channel activity. We also examined the interaction on TRPM2 channels, since the channel is expressed in endothelial cells and inhibited by Cu^2+^ [35,46]. Hcy had no significant effect on TRPM2, but it prevented the inhibitory effect of Cu^2+^ (Appendix A). These data indicate that the complexes of Hcy–copper or the charge of copper ions may be the determinant for their effects on ion channels.

### 3.4. No Effect of Monovalent Cu^+^ on TRPC Channel

To test the role of copper ion charges, we examined the effects of monovalent copper (I) compounds. As shown in Figure 4, the copper (I), (1,10-phenanthroline)bis(triphenylphosphine) copper (I) nitrate dichloromethane adduct, had no effect on TRPC4α and TRPC5 channel activity, but the divalent Cu^2+^ activated them (Figure 4A–C). Similarly, no effects of the monovalent copper, copper (I) 1-butanethiolate), and copper (I) tetrakis(acetonitrile) copper(I) tetrafluoroborate) were observed on TRPC4α channels (Appendix A). These data suggest that the divalent copper ions are essential for TRPC channel activation, but there are no effects for monovalent Cu^+^ ions. In addition, Se^2+^ with antioxidant properties had no effect on TRPC4α channels (Figure 4D–F), suggesting that the TRPC channel has metal ion specificity. The conversion from divalent to monovalent copper ions under oxidative stress conditions could be an important part of endogenous regulators for TRPC4 and TRPC5 channel activity.

### 3.5. Extracellular Activation of Cu^2+^ on TRPC4 and 5 Channels

Whole-cell patch recordings were performed using a pipette solution containing 10 μM Cu^2+^. The activation of the TRPC4 current by the intracellular Cu^2+^ application did not happen after the whole-cell configuration was formed for more than 5 min; however, bath perfusion with 10 µM Cu^2+^ significantly activated the current of TRPC4α with typical IV curves (Figure 5A). A similar effect on TRPC5 was observed (Figure 5B). We also performed outside-out excised membrane patches and the stimulating effects on TRPC4 and TRPC5 currents by Cu^2+^ were significant after the external surface exposure to Cu^2+^ by bath perfusion (Figure 5C,D). These data suggest that the action site for Cu^2+^ is extracellularly located.

### 3.6. Amino acid Residues of TRPC4 Involved in Copper Activation

To identify the action site of channel activation by Cu^2+^, we substituted the negatively charged glutamic acids (E) at the position of E542, E543, and E555 with the uncharged amino acid glutamine (Q); the cysteine (C554) with tryptophan (W); and the positively charged lysine (K) with the negatively charged glutamic acid (E) in the putative extracellular loops between the S5 and S6 domain of TRPC4α (Figure 6). The mutants of E542Q/E543Q, E555Q, C554W, and K556E did not affect the membrane trafficking of the channel proteins; however, the mutants of E542Q/E543Q and C554W caused resistance to Cu^2+^, but these mutants did not alter the sensitivity to trypsin, since trypsin is assumed to be an intracellular signalling process through GPCR activation (Figure 6). The mutants E555Q and K556E did not significantly change the effect of copper activation. These data indicate that negatively charged glutamic acids and the cysteine residue in the third extracellular loop are functional targets for divalent copper.

### 3.7. TRPC and Homocysteine-Copper Complexes in the Regulation of Endothelial Cell Proliferation

The blocking of TRPC channels has been shown to inhibit cell proliferation by us and others [27,42,47]. Here we further demonstrated the roles of TRPCs in the endothelial cells from macrovasculature. The proliferation of HAECs was significantly inhibited by specific pore-blocking TRPC antibodies (Figure 7A), which was consistent with the nonselective blocker 2-APB (Figure 7B). The over-expression of TRPC1 or TRPC4 promoted proliferation (Figure 7C), suggesting the significant contribution of TRPC channel activity to endothelial cell proliferation. However, Hcy inhibited the proliferation of HAECs but increased the proliferation of HUVECs. The pro-proliferative effect was more pronounced in the culture medium omitting cysteine and methionine (Figure 7D), or in the T-REx cells overexpressing Hcy-sensitive TRPC5 channels (Appendix A). On the other hand, divalent copper had no significant effect on the proliferation of HAECs but significantly reduced the proliferation of HUVECs and the HUVEC-derived cell line EA.hy926 (Figure 7E). Combined incubation with Hcy and Cu^2+^ showed inhibitory effects at low concentrations of copper but stimulatory effects at a high concentration (100 µM Cu^2+^) (Figure 7F), which exhibited significant differences from the groups treated with Hcy alone. These data suggest that the sensitivity to Hcy and Cu^2+^ may rely on the types of vascular endothelial cells and the ratio of Hcy and copper complexes.

### 3.8. Hcy–Copper Complexes in the Regulation of Cell Migration and Angiogenesis

TRPC channels are involved in cell migration and angiogenesis [26,50,51], so we observed the effects of Hcy and copper on endothelial cell migration and angiogenesis. Using a linear wound assay, the number of migrated cells was seen to be significantly reduced after treatment with Hcy (Figure 8A,B). Angiogenesis was examined using the extracellular matrix (ECM) gel and Matrigel assays. The score of angiogenesis in the ECM gel and the tube formation in the Matrigel were significantly inhibited by Hcy (Figure 8C–G). However, the addition of Cu^2+^ in the culture medium alleviated the inhibitory effects of Hcy on endothelial cell tube formation and angiogenesis, suggesting that endothelial cell mobility and angiogenesis are regulated by the complexes of homocysteine and copper. Taken together, regulation by Hcy and copper complexes via TRPC4/TRPC5 channels could be regarded as a new mechanism to control endothelial function.

## 4. Discussion

Our data show that Hcy can increase Ca^2+^ influx in HAECs. The increase is mediated by the opening of TRPC4 and TRPC5 channels. Divalent copper acts as a non-selective activator of TRPC4/5 channels. The channel activation by divalent copper is regulated by Hcy. The charge of copper ions is critical for TRPC channel opening because monovalent copper (I) shows no significant effect on TRPC channel activity. We also explored the action site for divalent copper using excised membrane patches and site mutagenesis. The cysteine (C^554^) and glutamic acids (E^542^ and E^543^) in the third extracellular loop of TRPC4α are responsible for copper activation. Moreover, we showed that copper and Hcy are essential regulators for endothelial cell proliferation, migration, and angiogenesis. Divalent copper seems to counteract the effect of Hcy on proliferation and angiogenesis which suggests the importance of the Hcy–copper interaction in causing endothelium dysfunction and atherosclerosis. The regulation of TRPC channels is the sought-after underlying mechanism for the pathogenesis of patients with hyperhomocysteinemia.

The effect of Hcy on intracellular [Ca^2+^]_i_ is still unclear in endothelial cells, although there are several reports showing that Hcy increases the Ca^2+^ influx in human platelets [52], cultured vascular smooth muscle cells [23], podocytes [53], and neurons [24,54]. Here, we found that Hcy increased the Ca^2+^ influx in HAECs which is mediated by the activation of TRPC4 and TRPC5. The blocking of voltage-gated Ca^2+^ channels and NMDA receptors was unable to prevent the Hcy-induced Ca^2+^ influx, suggesting that the Hcy-induced Ca^2+^ entry pathway is not through the voltage-gated channel or the ligand-gated NMDA receptor channel in vascular endothelial cells. In addition, the Hcy-induced intracellular Ca^2+^ increase has been linked to ER calcium release via the homocysteine-inducible ER stress protein [55]; however, Hcy-induced Ca^2+^ influx also happened in the cells acutely treated with SERCA blocker TG which suggests that main pathways of Ca^2+^ influx are across the plasma membrane rather than the intracellular Ca^2+^ release from the ER. The effect of Hcy on store-operated channels or ORAI channels is unknown, but high concentrations (≥100 µM) of Hcy may inhibit the store-operated Ca^2+^ influx [56]. Hcy also inhibits BK_Ca_ and thus depolarises the membrane potential and increases the vascular tone [57]. This action may explain the diverse responses in vascular tone or [Ca^2+^]_i_ observed in some cell types [58,59]. The N-methyl-D-aspartate (NMDA) receptor activation by Hcy could also be a mechanism for Ca^2+^ influx in the nervous system [24] but this mechanism may be less significant in vascular endothelial cells.

Homocysteine contains sulphuric residues so its toxic effect has been attributed to redox homeostasis, such as the production of different reactive oxygen species (ROS), thus leading to the oxidation of low-density lipoprotein [20]. Cellular oxidative stress including ER stress has also been proposed for Hcy pathophysiology [19]; the increased ROS production activates ROS-sensitive Ca^2+^ channels. In addition, we demonstrated that the TRPC5 channel is a redox-sensitive channel that can be activated by thioredoxin and reducing agents [37] and mercury compounds [41]. Here, we found that TRPC4 and TRPC5 channel activities can be enhanced by Hcy, especially when the channels are opened by lanthanides. TRPM2 is also a redox-sensitive channel; however, Hcy itself had no acute effect on TRPM2 but significantly regulated the effect of Cu^2+^ on the TRPM2 channel. Chronic exposure to Hcy may change gene expressions, through Ca^2+^ channels and ROS signalling molecules [53,60], but we did not observe such gene expression in this study.

The total Hcy level in the blood is determined by both genetic and environmental factors and is typically maintained at a normal range (2–14 µM). Vitamin deficiencies, in particular folate acid and vitamins B6 and B12, appear to be the most common causes of elevated Hcy [61]. A supplement of folic acid alone or with vitamin B12 or B6 can help to lower Hcy levels, but it is still uncertain how effective this will be in the prevention of cardiovascular disease or Hcy-related diseases. It has been demonstrated that both Hcy and copper are increased in diseased vessels and diabetic patients; however, the question of how Hcy interacts with copper and causes occlusive diseases remains unanswered. Here, we show for the first time that copper can interact with Hcy, controlling TRPC channel activity, thus changing intracellular Ca^2+^ signalling, and subsequently the endothelial function. This mechanism gives a new understanding of the two factors in the pathogenesis of cardiovascular diseases. Too low or too high concentrations of copper are detrimental, but we have demonstrated that the charge of copper ions could be more important than the copper concentration. Although treatment with a divalent-copper-selective chelator, triethylenetetramine (TETA), to lower the copper in the body may improve the cardiac structure and function in patients and rats with diabetic cardiomyopathy [34], a more precise clinical trial is needed, especially regarding the charge of copper ions and consideration of the redox environment in the body.

The inhibition of TRPCs shows anti-proliferative effects while the activation of TRPC channels shows proliferative effects in vascular endothelial cells, which is consistent with the observations in other cell types [26,42]. However, different types of endothelial cells may show differences, such as the HAECs showing inhibitory characteristics and the HUVECs showing pro-proliferative characteristics. This could be related to the predominance of Hcy-sensitive channels. In patients with hyperhomocysteinemia, neointimal hyperplasia in small vessels is evident [1].

In summary, we revealed a new mechanism of Hcy and copper and their interplay with TRPC channels in endothelial cells. This new concept could be extended to other cell types since many diseases are related to Hcy and copper and Hcy is associated with all-cause mortality. The findings suggest the importance of copper ion charges in the pathogenesis of vascular disorders, particularly in patients with increased homocysteine levels, and may also provide an alternative explanation for why Hcy-lowering therapy is not very significant in clinical trials and how Hcy-copper complexes could be the determinants.

## Figures and Tables

**Figure 1 biomolecules-13-00952-f001:**
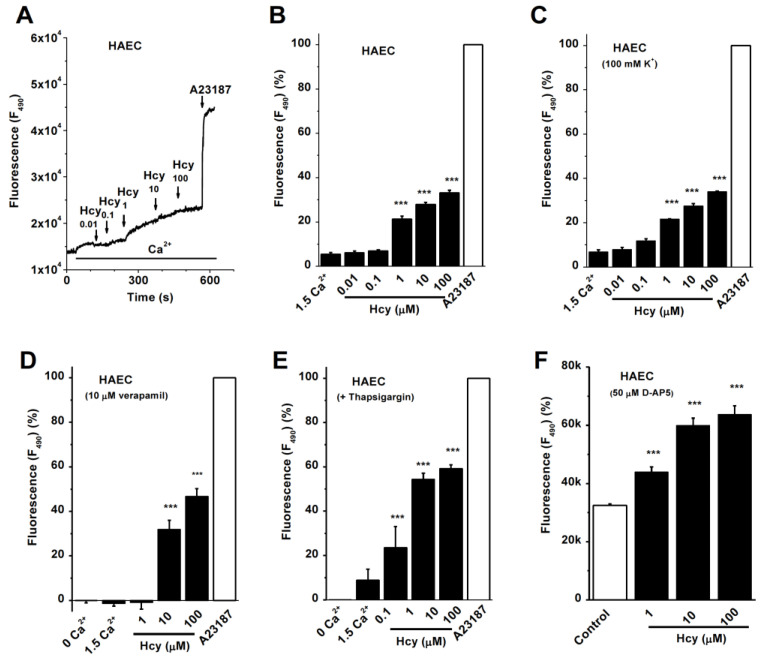
Effect of Hcy on Ca^2+^ influx in HAECs. Ca^2+^ influx was measured using Fluo-3 AM. (**A**) Example of Hcy on Ca^2+^ influx. Hcy was added accumulatedly and followed by calcium ionophore A23187 (2 µM). (**B**) The mean ± s.e.m. for the effect of Hcy. (**C**) Effect of Hcy under the bath solution with 100 mM K^+^. (**D**) Response to Hcy after blocking the voltage-gated Ca^2+^ channel with 10 µM verapamil. (**E**) Thapsigargin (2 µM) was added to block the SERCA. (**F**) NMDA antagonist 5-AP (50 µM) added. The ANOVA test was used and *n* = 6–8 for each experiment. *** *p* < 0.001.

**Figure 2 biomolecules-13-00952-f002:**
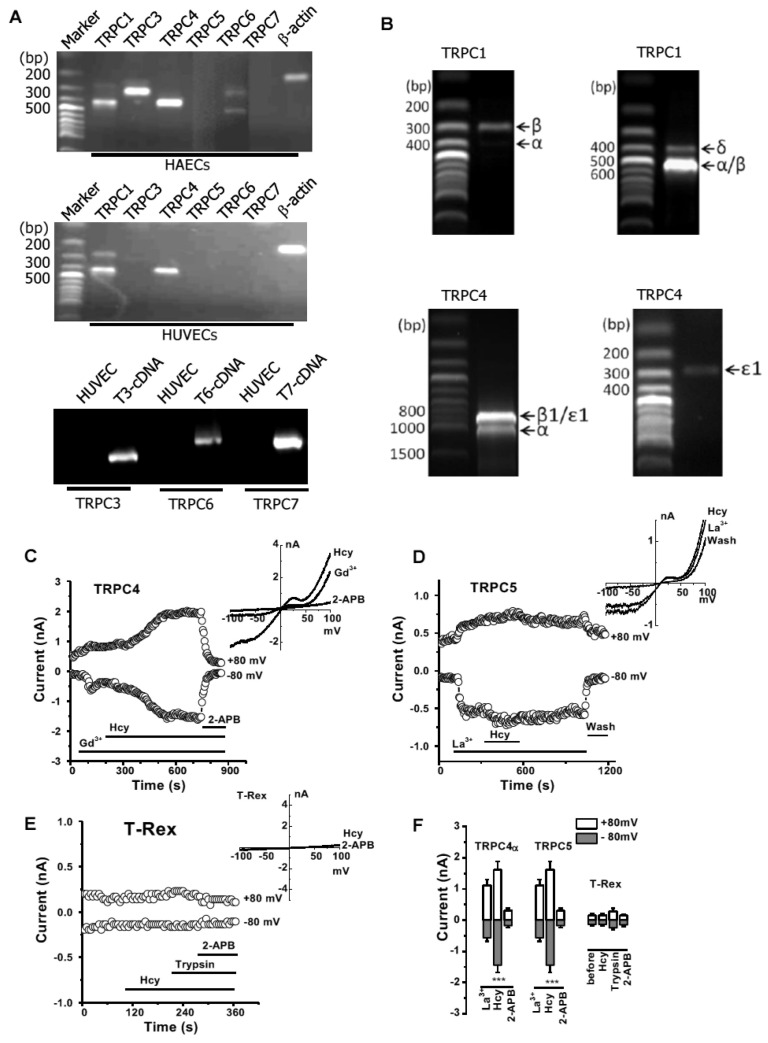
Hcy-induced Ca^2+^ influx through TRPC4 and TRPC5 channels in endothelial cells. (**A**) mRNAs of TRPCs in vascular endothelial cells (HAECs and HUVECs). The plasmid cDNAs for TRPC3, 6, and 7 were used as positive controls. (**B**) Detection of TRPC1 and TRPC4 spliced variants in HAECs. The PCR primers and the corresponding size of amplicons were given in our previous reports [42]. (**C**) TRPC4 current recorded in HEK293 T-REx cells inducibly overexpressing TRPC4α channels and the effect of Hcy (100 µM). (**D**) Current for induced TRPC5 cells. (**E**) Non-induced T-REx cell as control. (**F**) The mean ± s.e.m. measured at ±80 mV after exposure to each compound. *n* = 5–6 for each group. *** *p* < 0.001 compared with La^3+^ treatment measured at ±80 mV.

**Figure 3 biomolecules-13-00952-f003:**
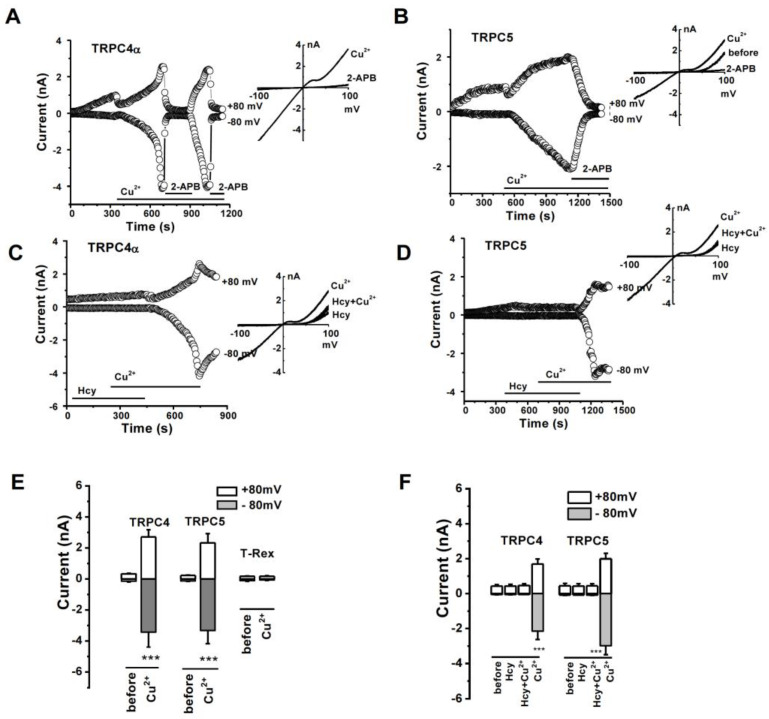
TRPC channel activated by Cu^2+^ and counteracted by Hcy. (**A**,**B**) Representative time course and IV curve for TRPC4 and TRPC5 activated by Cu^2+^. 2-APB (100 µM) as a control channel blocker. (**C**,**D**) TRPC4 and TRPC5 currents after perfusion with 100 µM Hcy, the addition of 10 µM Cu^2+^, and the washout of Hcy. (**E**) The mean ± s.e.m. data for the effect of Cu^2+^ (*n* = 6–8. *** *p* < 0.01). (**F**) The mean ± s.e.m. data for Hcy plus Cu^2+^ (*n* = 5–6. *** *p* < 0.001).

**Figure 4 biomolecules-13-00952-f004:**
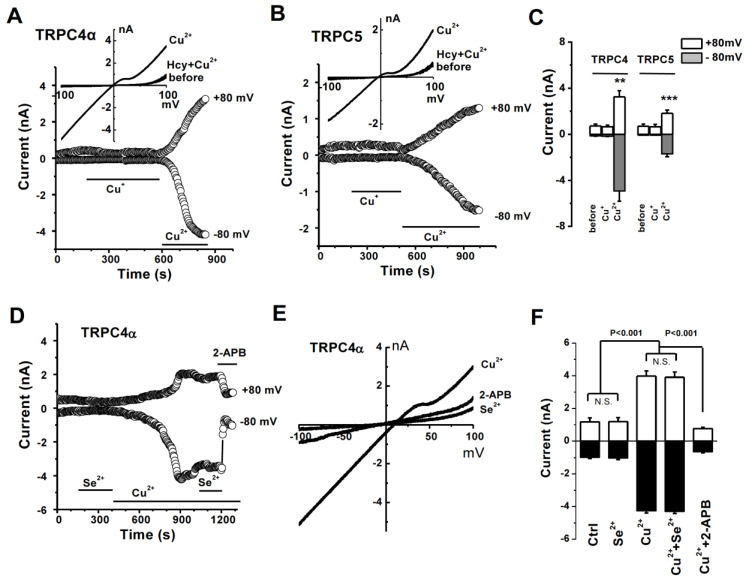
Monovalent copper (Cu^+^) had no effect on TRPC channels. (**A**) TRPC4 cells were perfused with 10 µM monovalent copper ((1, 10-phenanthroline), bis (triphenylphosphine) copper (I) nitrate dichloromethane adduct), and then 10 µM divalent Cu^2+^. (**B**) Similar to (**A**) but TRPC5 cells were used. (**C**) The mean ± s.e.m. data measured at ± 80 mV after perfusion with Cu^+^ and Cu^2+^. *n* = 5–7 for each group, ** *p* < 0.01 and *** *p* < 0.001. (**D**) Effect of sodium selenite on TRPC4 current. (**E**) IV curves for (**D**). (**F**) The mean ± s.e.m. data for the effect of Se^2+^ and Cu^2+^ on TRPC4 current.

**Figure 5 biomolecules-13-00952-f005:**
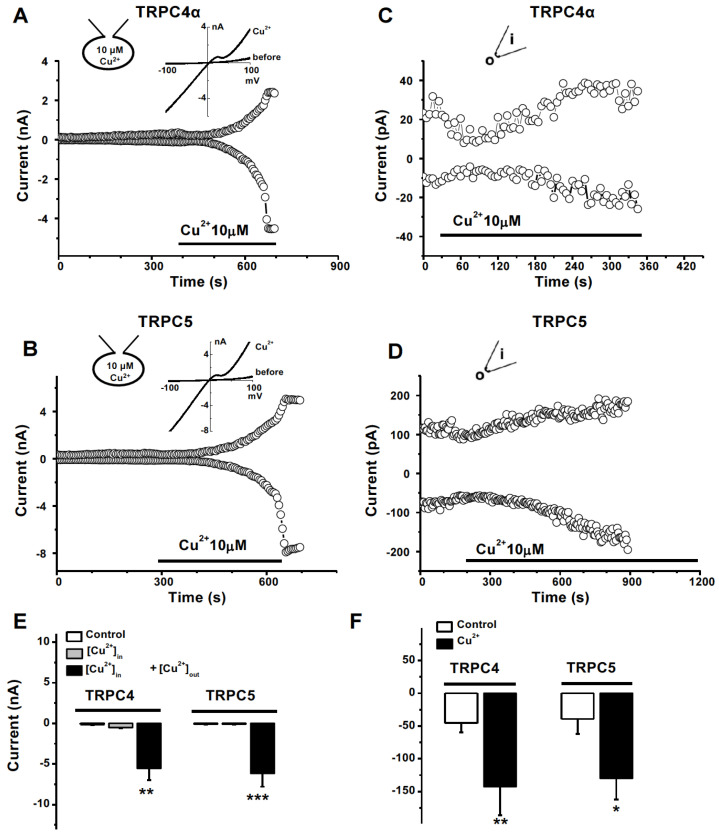
Extracellular effect of Cu^2+^ on TRPC4 and TRPC5 channels. (**A**) A whole-cell patch was recorded in the HEK293 T-REx cells overexpressing TRPC4α with a pipette solution containing 10 µM Cu^2+^ (*n* = 4 for each group). (**B**) Same as (**A**) but cells overexpressing TRPC5 cells were used. (**C**) Example of outside-out patches showing the effect of Cu^2+^ on TRPC4α. (**D**) Outside-out patches for TRPC5 channels. (**E**) The mean ± s.e.m. for (**A**) and (**B**) (*n* = 4). (**F**) The mean ± s.e.m. for (**C**,**D**) (*n* = 4). * *p* < 0.05, ** *p* < 0.01, and *** *p* < 0.001.

**Figure 6 biomolecules-13-00952-f006:**
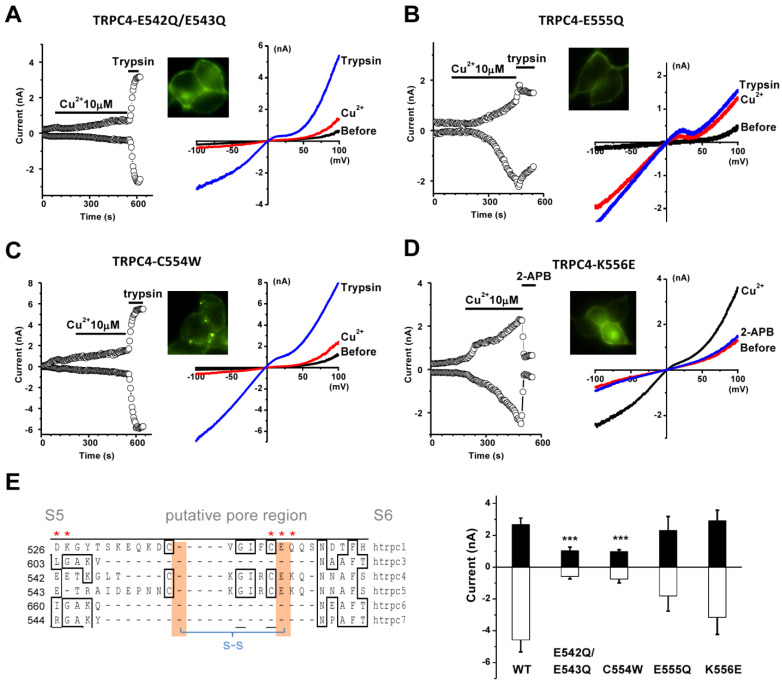
Identification of amino acids involved in channel activation by Cu^2+^. The mutants of TRPC4α tagged with EYFP were made by site-mutagenesis and membrane localisation was examined using a fluorescent microscope. (**A**) The double glutamic acid mutants (TRPC4-E542Q/E543Q) showed the loss of channel activation by Cu^2+^, but the robust current through the mutant channel can also be activated by trypsin (2 nM). (**B**) The TRPC4-E555Q mutant was activated by Cu^2+^. (**C**) Less sensitivity to Cu^2+^ for the cysteine mutant (TRPC4-C555W). (**D**) Glysine at the position of 556 substituted with glutamic acid (TRPC4-K556E). (**E**) Amino acid alignment of the transmembrane region (S5-S6) for TRPCs (red asterisks indicate residues subject to mutagenesis) and the mean ± s.e.m. data showing the amplitude of currents corresponding to the mutants and the wild-type control after perfusion with Cu^2+^ (*n* = 8). *** *p* < 0.001.

**Figure 7 biomolecules-13-00952-f007:**
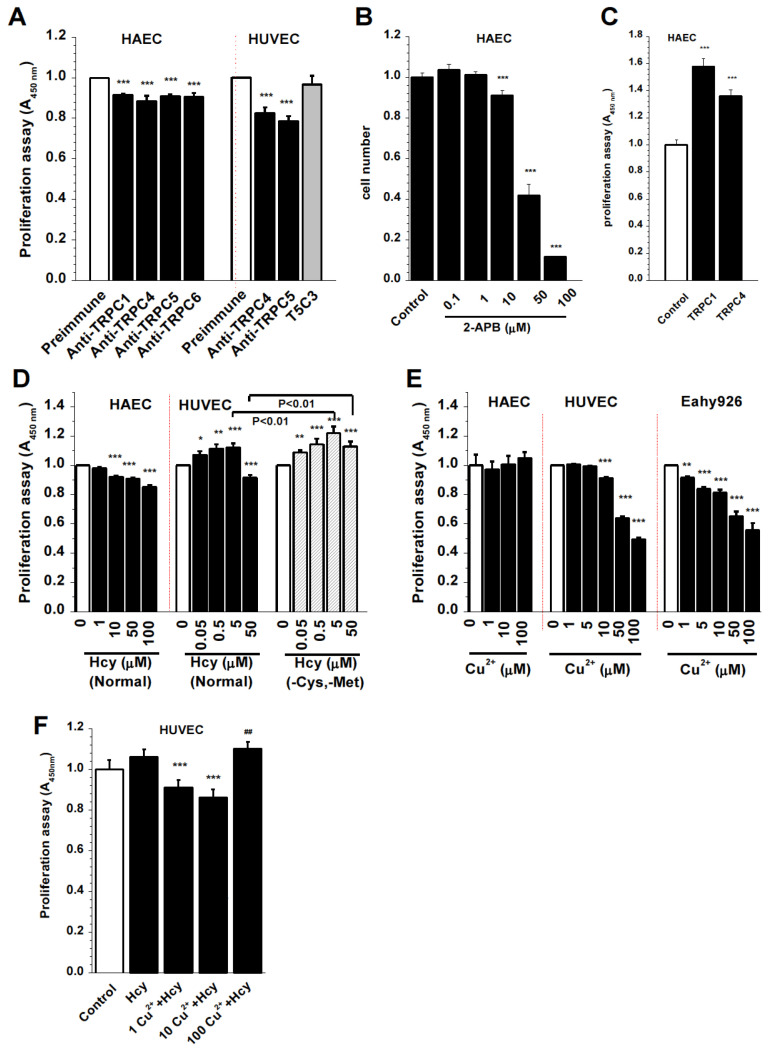
Endothelial cell proliferation regulated by TRPC channels and the effects of Hcy and copper. Cell proliferation was assayed by a WST-1 kit and absorbance was measured at a wavelength of 450 nm. (**A**) Endothelial cells were incubated with the pore-blocking TRPC antibodies [28,42,48] for 24 h. The TRPC5 antibody targeting the C-terminal (T5C3) and preimmune serum (Preimmune) were used as controls. (**B**) 2-APB. (**C**) HAEC cells transfected with plasmid cDNAs for TRPC1 and TRPC4 using the electroporation method [49]. (**D**) Effect of Hcy on HAECs and HUVECs. (**E**) Effect of Cu^2+^ on HAEC, HUVEC, and the HUVEC-derived cell line Eahy926. (**F**) Combined effect of Hcy (10 µM) and Cu^2+^. *n* = 8 for each group, * *p* < 0.05, ** *p* < 0.01, and *** *p* < 0.001, ^##^, not significant.

**Figure 8 biomolecules-13-00952-f008:**
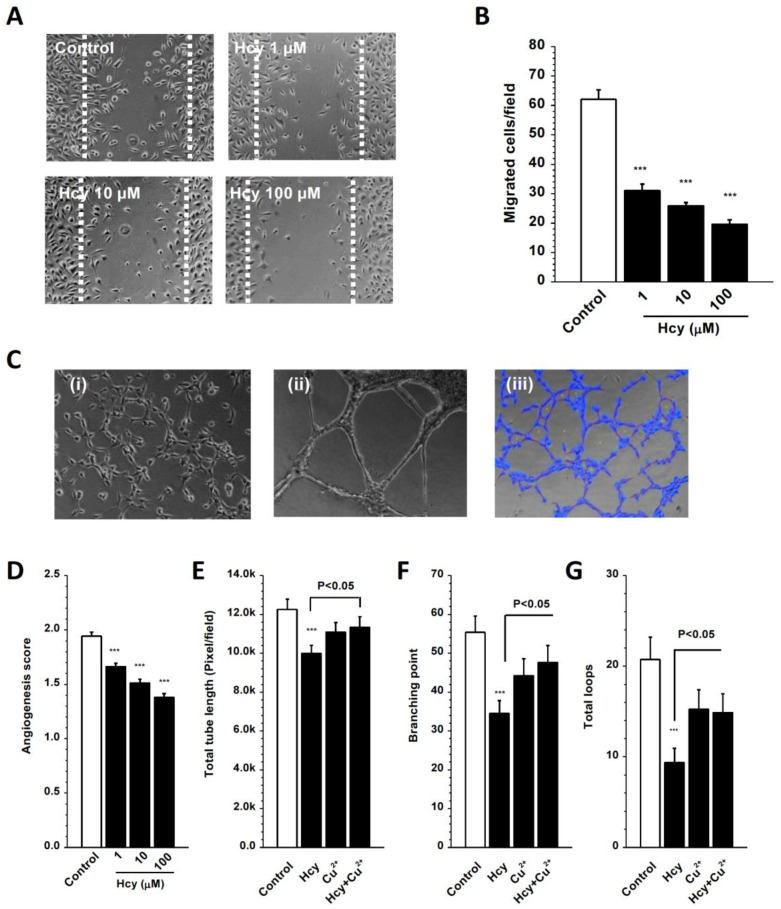
Endothelial cell migration and angiogenesis regulated by Hcy and Cu^2+^ complexes. (**A**) Example of endothelial cell migration using a linear wound assay. (**B**) Effect of Hcy on cell migration after 24 h of incubation. (**C**) Example of angiogenesis using ECM gel (**i**) and Matrigel for HUVEC (**ii**) and Eahy926 cells (**iii**). (**D**) The mean ± s.e.m. showing the effect of Hcy on angiogenesis (*n* = 40–60 imaging fields from six cell culture dishes for each group). (**E**–**G**) Effect of Hcy (10 µM) and Cu^2+^ (10 µM) on endothelial cell tube formation. *n* = 6 for each group. The number of loops, branching, and total length of tubes were analysed by software. *** *p* < 0.001.

## Data Availability

The data supporting this study are available from the corresponding authors upon reasonable request.

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
