# Peer review of "Ca2+ Influx through TRPC Channels Is Regulated by Homocysteine–Copper Complexes"

_biomolecules, 2023, doi:10.3390/biom13060952_

Round 1

Reviewer 1 Report

The authors report on their studies on the effects of homocysteine and copper on activation of recombinant TRPCs expressed in HEK293 cells and on endothelial cells. They show that Hyc increases calcium influx and that channel activation is regulated by extracellular Cu2+, interacting with the extracellular pore region of TRPCs. Some points need to be considered: 

lines 73-77:Please provide more details on the expression vectors for the different TRPC isoforms and on the transfection protocol.

line 83, please delete "in" in the sentence: "cultured in in endothelial cell growth medium"

line 102: please provide the composition of standard bath solution

line 110: please delete "containing" 

line 145 and figure 1D: the authors show the effect of Hcy on the intracellular calcium concentration of HAEC cells. Cells were treated with 100 mM K+ or with verapamil to evaluate the role of voltage-dependent Ca2+ channels . The authors comment that treatment with K+ or verapamil did not change the effect of Hcy. However figure 1D shows that the effect of 1 micro molar Hyc is significantly different in the absence or in the presence of verapamil. Can the author comment on this? Do they exclude a non specific effect of verapamil on Hcy-induced calcium entry?

Line 152 and 155: figure 1E and 1F have been exchanged in the text , Fig1E refers to the effect of thapsigargin and Fig. 1F to that of D-AP5

Figure 3C: the authors state that treatment with Hcy prevented TRPC4a activation by Cu2+; however a TRPC4a activation can be observed, although delayed. May the authors comment on this point?

Figure 8: the authors state that the inhibitory effect of Hcy on angiogenesis (tube lenght, branching point and total loops) can be partially reversed by Cu2+. Is there a statistically significant  difference between the effect of Cu2+ alone and Hcy+Cu2+ on angiogenesis. In other words, can the effect on abiogenesis be explained by Cu2+ alone, regardless the presence of Hcy?

line 317, please replace showing with show

Line 337-341: the authors state that: "In addition, Hcy-induced intracellular Ca2+ increase has been linked to ER calcium release via homocysteine-inducible ER stress protein; however, the Hcy induced Ca2+ influx also happened in the cells acutely treated with SERCA blocker TG, which suggests that main pathways of Ca2+ influx are across the plasma membrane, rather than the intracellular Ca2+ release from ER". The authors may also consider activation of Store Operated calcium Entry (SOCE), induced by treatment with TG as a mechanism regulating calcium influx. A comment on this point should be added in the discussion.

English language needs some moderate editing

Reviewer 2 Report

The manuscript entitled "Ca2+ influx through TRPC channels is regulated by homocysteine-copper complexes" by Chen et al. describes the regulation of transient receptor potential TRPC5 and TRPC6 channels by homocysteine and Cu2+. Homocysteine interfered with Cu2+-induced TRP channel activation. Mutation analysis identified specific glutamic acid and cysteine residues within the TRPC4 channel responsible for the effect of homocysteine and Cu2+. Further experiments addressed the effect of homocysteine on endothelial proliferation, migration and angiogenesis. The authors conclude that homocysteine/Cu2+ represents a pair of endogenous regulators and that Cu2+-chelating therapy may have potential benefit in the treatment of hyperhomocysteinemia. 

A number of functional and technical questions need to be addressed.

1. It is not clear to me why the authors used La3+ and Gd3+ for TRP channel activation and not TRP-specific compounds such as (-)-Englerin or BTDAzo.

2. The authors used TRPC-specific pore-blocking antibodies for their experiments. In addition, the compound 2-ABP was used in several experiments. 2-ABP is a very unspecific channel blocker that impairs the activities of numerous TRP channels and also inhibits L-type voltage gated Ca2+channels. It also inhibits signal transduction via Gaq-coupled receptors. Therefore, this compound is no longer useful and should be replaced by more specific compounds (e.g., HC-070 or clemizole for TRPC4 and TRPC5) or by using genetic tools. 

3. The experiments concerning the regulation of proliferation are puzzling. The data presented in Fig. 7A show that incubation with pore-blocking antibodies directed against TRPC1, TRPC4, TRPC5 and TRPC6 have a small, but apparently significant effect on proliferation of HAEC cells, even though the cells do not express TRPC5 and only slightly express TRPC6. Why do pore-blocking antibodies against TRPC5 have an effect on HAEC cells that do not express TRPC5? HUVEC cells also do not express TRPC5, but the pore-blocking antibodies slightly decrease proliferation. These experiments were performed without stimulation of the TRPC channels. What is the effect of channel stimulation on cell proliferation? In Fig. 7C, overexpression of TRPC1 and TRPC4 in HAEC cells stimulate cell proliferation. The cells already endogenously express these TRP channels. Therefore, I wonder if the authors think that TRPC1 and TRPC4 channels regulate the proliferation of HAEC cells dependent on their concentration and independent of their activation.

4. The experiments on cell migration and angiogenesis were also performed without stimulation of TRP channels. Do the authors consider that activation of TRPC channels plays no role in the regulation of cell migration and angiogenesis?

Round 2

Reviewer 2 Report

no further comments